# Scleral Thickness as a Risk Factor for Central Serous Chorioretinopathy and Pachychoroid Neovasculopathy

**DOI:** 10.3390/jcm12093102

**Published:** 2023-04-24

**Authors:** Leonie F. Keidel, Benedikt Schworm, Julian Langer, Nikolaus Luft, Tina Herold, Felix Hagenau, Julian E. Klaas, Siegfried G. Priglinger, Jakob Siedlecki

**Affiliations:** Department of Ophthalmology, Ludwig-Maximilians-University, 80336 Munich, Germany

**Keywords:** pachychoroid, pachysclera, swept-source optical coherence tomography, anterior scleral thickness, central serous chorioretinopathy, pachychoroid neovascularisation

## Abstract

In the pathophysiology of central serous chorioretinopathy (CSC), scleral changes inducing increased venous outflow resistance are hypothesized to be involved. This work aims to investigate anterior scleral thickness (AST) as a risk factor for pachychoroid disorders. A randomized prospective case-control study was performed at the Ludwig Maximilians University, Department of Ophthalmology. In patients with CSC or pachychoroid neovasculopathy (PNV) and in an age- and refraction-matched control group, swept source optical coherence tomography (SS-OCT) was used to measure anterior scleral thickness (AST). Subfoveal choroidal thickness (SFCT) was assessed using enhanced depth imaging OCT (EDI-OCT). In total, 46 eyes of 46 patients were included in this study, with 23 eyes in the CSC/PNV and 23 eyes in the control group. A significantly higher AST was found in the CSC/PNV compared with the control group (403.5 ± 68.6 (278 to 619) vs. 362.5 ± 62.6 (218 to 498) µm; *p* = 0.028). Moreover, the CSC/PNV group showed a higher SFCT (392.8 ± 92.8 (191–523) vs. 330.95 ± 116.5 (167–609) µm, *p* = 0.004). Compared with the age- and refraction-matched controls, patients with CSC and PNV showed a significantly thicker anterior sclera. Scleral thickness might contribute to the venous overload hypothesized to induce pachychoroid phenotypes.

## 1. Introduction

Pachychoroid disorders represent a novel entity of macular degeneration and include pachychoroid pigment epitheliopathy (PPE), central serous chorioretinopathy (CSC), pachychoroid neovasculopathy (PNV), and pachychoroid aneurysmal type 1 choroidal neovascularization/polypoidal choroidal vasculopathy (PAT1/PCV) [1]. The pathogenesis of the pachychoroid disease spectrum manifests as a congested and thickened choroid and is currently thought to involve compression of the vortex veins and secondary reflux into the choroidal venous system [2]. In a study conducted by Matsumoto and colleagues, pachychoroid features could be induced in mice by suturing of a vortex vein, thus reducing venous efflux from the choroid [3].

Kishi and colleagues recently described a frequent overlap of areas of vortex vein dilatation and areas of reduced perfusion of the choriocapillaris in en-face optical coherence tomography (OCT) imaging in eyes with CSC [4]. They concluded that in CSC, compression of the vortex veins can lead to backflow into the venous system of the choroid and, secondarily, to anterograde filling delay of the choriocapillaris with the formation of ischemic areas [5]. This thesis is supported by the fact that in eyes with CSC, chronic overload of the venous system was observed compared with a control collective. An excess in the outflow areas of the vortex veins was seen in response to the formation of intervenous anastomosis and was significantly increased in eyes with CSC [6]. Thus, the term “venous overload choroidopathy” was recently introduced by Spaide and colleagues [7].

The exact pathophysiology of the pachychoroid spectrum has yet to be fully explained, especially concerning the influence of the sclera on vortex vein drainage. Therefore, the aim of this study was to investigate whether scleral changes, especially scleral thickness, represent an ocular risk factor in CSC and PNV. This study is also the first to investigate scleral thickness in eyes with PNV.

## 2. Materials and Methods

This prospective case-control study enrolled 23 consecutive eyes presenting with CSC or PNV at the Hospital of the Ludwig Maximilians University Munich, Department of Ophthalmology, Germany, from March 2022 to June 2022. They were matched by age and spherical equivalent (SE) to a randomly recruited normal control group (23 eyes) from the same time span. Institutional review board approval was obtained for this prospective case-control study (Identifier: 22-0027). The study adhered to the tenets of the Declaration of Helsinki. All patients provided written consent prior to any study-related procedure.

Epidemiological data were obtained from each patient, including age, gender, previous ocular comorbidities or procedures, and objective refraction-based Snellen chart visual acuity, which was later converted to logMAR for analysis. Patients’ spherical equivalents and spheres were obtained using the auto refractometer Nidek AR-1s (Oculus GmbH, Wetzlar, Germany).

### 2.1. Imaging

Imaging included the acquisition of a standard volume scan centered on the macula consisting of 49 equally spaced B-scans covering 20 × 20 degrees centered on the fovea (on Spectralis HRA + OCT, Heidelberg Engineering, Heidelberg, Germany) [8]. Subfoveal choroidal thickness (SFCT) was assessed by manual measurements of the distance between Bruch’s membrane and the transition zone between the choroid and the sclera underneath the foveola.

CSC is defined as a serous retinal detachment with or without serous pigment epithelial detachment (PED) atop a thickened choroid with typical pachyvessels in Haller’s layer and thinning of the choriocapillaris [9]. Chronic CSC is present when persistent subretinal fluid for a duration of 6 months can be shown. For the diagnosis of PNV, choroidal neovascularization (CNV) formation atop a thickened choroid had to be visualized using OCT. Fluorescein and indocyanine green angiography (ICGA) had to reveal a staining plaque on ICGA and flow within the CNV had to be seen using OCT angiography [9].

Anterior scleral thickness (AST) was measured using a swept source OCT (SS-OCT) equipped with an anterior segment module (Dri OCT Triton, Topcon, Tokyo, Japan). Images of the temporal sclera were captured during 45° nasal gaze position using a 6 mm scan. Scleral thickness was measured manually using the built-in caliper function below the insertion point of the lateral rectus muscle. The anterior scleral stroma extends from the rectus muscle/sclera junction to the sclera/choroid junction and was measured perpendicularly between the junctions (Figure 1).

### 2.2. Statistical Analysis

All data were gathered and analyzed in Microsoft Excel spreadsheets (Version 16.23 for Mac; Microsoft, Redmond, WA, USA). Statistical analysis was performed in SPSS Statistics 26 (IBM Germany GmbH, Ehningen, Germany). Statistical significance was defined as *p* < 0.05. The Kolmogorov−Smirnov tests were used to test for normal distribution. Inter-group differences were analyzed using the independent two-tailed Student t-test and the Mann−Whitney U Test.

## 3. Results

### Patient Characteristics

The CSC/PNV group included 23 eyes of 23 patients with a male to female ratio of 15:8, a mean age of 51.5 ± 8.0 (36–65) years, and a spherical equivalent of 0.60 ± 2.23 (−5.5–4.6). Ten patients had a myopic refraction (2 PNV patients) and thirteen patients had a hyperopic refraction. CSC/PNV was idiopathic in 19 patients (iCSC), whereas in 4 patients, a steroid-induced CSC (sCSC) was present. There were 2 patients with acute CSC, 12 patients with chronic CSC, and 9 patients with pachychoroid neovasculopathy. The PNV subgroup included these nine eyes of nine patients with a mean age of 57.4 ± 5.4 (50–65) and a spherical equivalent of 1.33 ± 1.71 (−1–4.6). At the time of enrollment, seven patients with chronic CSC had been exclusively treated with 25 mg of oral spironolactone twice daily for two to six months, one patient had received both spironolactone and subthreshold laser treatment (EPM, endpoint management) [10], one patient had received solely EPM treatment, one patient had received photodynamic therapy, and two patients had not received any treatment yet. From the PNV subgroup, all patients had received 3–37 (mean 14 ± 13.5) anti-VEGF injections, one patient had received spironolactone treatment beforehand, and one patient had also received photodynamic therapy twice. From the two patients with acute CSC, one patient had received focal micropulse laser treatment, the other had not been treated yet. A detailed overview of the patients’ treatment history can be found in Table 1.

The control group consisted of 23 eyes from 23 subjects, with a male to female ratio of 14:9, a mean age of 47.0 ± 12.9 (30–83) years, and a spherical equivalent of −0.93 ± 2.45 (−4.9–5.8). Fourteen patients had a myopic refraction, three were emmetropic, and six patients were hyperopic. Detailed patient characteristics of the CSC and the control group in a cross comparison can be found in Table 2.

There was no difference in mean age (*p* = 0.122) and spherical equivalent (*p* = 0.441) between the CSC/PNV group and the control group, as well as between the PNV subgroup and the control group (*p* = 0.194, *p* = 0.347).

A significantly higher temporal AST was found in the CSC/PNV (403.5 ± 68.6 (278–619) µm) compared with the control group (362.5 ± 62.6 (218–493) µm; *p* = 0.028) (Figure 2, Table 2). Comparing the PNV subgroup only, the anterior scleral thickness continued to be significantly higher compared with the normal controls (403.6 ± 44.1 (341–472) µm; *p* = 0.024) (Figure 3). Moreover, the CSC/PNV group showed a significantly higher SFCT (392.8 ± 92.8 (191–523) µm) compared with the control group (330.95 ± 116.5 (167–609) µm, *p* = 0.04) (Table 2). AST did not show a significant positive correlation with SFCT in the CSC (R = −0.06, *p* = 0.80), nor in the control group (R = −0.23, *p* = 0.19). The CSC/PNV partner eyes all showed a pachychoroid (mean SFCT: 405.4 ± 120.7 (260–564) µm). There were eight eyes with PPE and two eyes with chronic CSC.

In the CSC collective and in the normal control group, the AST increased in thickness with the increase in hyperopia (R = 0.23, *p* = 0.30 and R = 0.21, *p* = 0.33), yet these findings were shown to be nonsignificant.

AST in patients with sCSC (341.8 ± 51.9 (278–392) µm) was remarkably low compared with those with iCSC (420.2 ± 64.6 (322–619) µm).

There was no significant difference in AST between the male and female gender in the CSC (403.7 ± 79.7 (278–619) µm vs. 411.8 ± 44.9 (347–472) µm, *p* = 0.75) and the normal control group (352.8 ± 49.8 (308–468) µm vs. 377.6 ± 79.6 (218–493) µm, *p* = 0.37).

Despite anti-VEGF treatment, the PNV subgroup did not show a significant thinning of AST compared with the CSC group (403.5 ± 46.2 (341–472) µm vs. 408.4 ± 81.4 (278–619) µm, *p* = 0.75).

## 4. Discussion

The present study revealed a significantly higher AST in the temporal segment in patients with CSC and pachychoroid neovasculopathy compared with the normal control group, matched to age and refraction.

CSC belongs to the “pachychoroid disease spectrum” [1]. Pachychoroid disorders are characterized by a thickening of the Haller’s layer of the choroid with a pachyvessel configuration and a thinning of the overlying Sattler’s layer. The thickening of the Haller’s layer resulting in increased hydrostatic pressure and thinning of the Sattler’s layer, resulting in ischemic areas that could lead to changes in the retinal pigment epithelium (RPE) (PPE stage, RPE mottling). Defects may develop between the RPE cells and, as a consequence, fluid may be emitted below the retina and diffuse leakage may occur (CSC stage) [11]. Hydrostatic pressure on the endothelial cells of the vessel wall can lead to remodeling of the vessel and a process called “arteriogenesis” [12,13]. This may result in pachychoroid neovasculopathy (PNV) [14,15,16].

To understand the pathophysiology of CSC, it is useful to look at the pathophysiology of uveal effusion syndrome (UES). This also presents with thickening of the choroid, sometimes to such an extent that the choroid detaches, and exudative retinal detachments occur in the periphery, as well as the central retina. In 2019, Jin and colleagues demonstrated specific pachychoroid features such as thinning of the Sattler’s layer and thickened Haller’s veins in a case with UES [17].

Physiologically, there is a balance between fluid flows in the choroid: the hydrostatic pressure of the capillaries is in equilibrium with the colloid osmotic pressure in the extravascular space. Colloids such as albumin, which escape the fenestrated capillaries, move out of the choroid through a transscleral fluid flow [18].

A thickened and rigid glycosaminoglycan-rich sclera is thought to be present in UES. Colloids are prevented from penetrating the sclera. As a consequence, retention of colloids in the choroidal interstitium may occur and fluid may subsequently accumulate, resulting in choroidal thickening [19].

In addition, it is assumed that a thick sclera leads to compression of the vortex veins draining the choroid, and thus causes a secondary backflow into the venous system of the choroid. The increased hydrostatic pressure in the choroidal capillaries aggravates the effect of extravascular fluid retention [20]. The choroid becomes thickened and—similar to CSC—there may be decompensation of the retinal pigment epithelium and exudation of fluid into the subretinal space. As scleral thickness increases with the decrease in axis length, an association between UES with short axial length has been described [21]. UES occurs predominantly in men.

As mentioned before, the pathogenesis of CSC is also currently thought to involve compression of the vortex veins and secondary reflux into the choroidal venous system. The term “overload choroidopathy” was thus recently introduced [7]. Kishi, Matsumoto, and colleagues recently described a frequent overlap of areas of vortex vein dilatation and areas of reduced perfusion of the choriocapillaris in en-face OCT imaging in eyes with CSC due to a retrograde flow into the choroidal veins and secondarily anterograde disturbed filling of the choriocapillaris with the development of ischemia in the choriocapillaris [4,22,23]. The resulting ischemia in the choriocapillaris, in combination with a thickened Haller’s layer, may ultimately lead to the development of the typical pachychoroid features, as described above. Longstanding vortex vein congestion may then result in the development of inter-vortex-venous anastomosis as a compensatory mechanism to the chronic overload of the venous system [23]. This theory is confirmed by the fact that CSC is also predominantly found in hyperopic, male eyes, congruent to UES [24,25]. Our data confirm the trend towards a thicker AST in hyperopia and decreasing AST in increasing myopia, recently described by Dhakal and colleagues [26]. In a recent important work by Sawaguchi, Koizumi et al., the mean AST in eyes with sCSC was shown to be significantly thinner than in the eyes with iCSC. Our findings go hand in hand with this work [27]. We did not observe any significant difference in scleral thickness between the two sexes, but at present there is still no consensus in the literature regarding this matter [28,29,30].

The significantly thickened anterior sclera may represent an ocular outflow resistance for the vortex veins in eyes with CSC, congruent with the pathophysiology of UES. Our data indicate that, consequently, the choroid may chronically thicken due to venous system overload, as seen by increased SFCT in the CSC/PNV group, and diseases from the pachychoroidal disease spectrum, such as CSC and PNV, may develop. This also provides new evidence that CSC and PNV form a continuum in the pachychoroid disease group.

The unique aspect of this study, compared with previous manuscripts [31,32], is the inclusion of eyes with a later stage of the pachychoroid disease spectrum [33], PNV, and that this subgroup alone also showed a significantly higher AST compared with the normal controls.

The fact that scleral thickness was measured away from the injection site of anti-VEGF therapy and that only an average of 14 anti-VEGF injections per patient were administered allowed us to include these eyes without the confounder of scleral thinning after repeated anti-VEGF therapy (>30 injections), as described by Zinkernagel and colleagues [34].

Using SS-OCT, an easily feasible, repeatable, and highly accurate tool (wavelength of 1050 nm, axial scan rate of 100,000 A-scans/second), measurement of the anterior sclera is possible [35]. SS-OCT is described as providing slightly more accurate scans than the recently used ultrasound B scan [36], with almost in vivo “histology” images of the anterior segment [37].

Limitations of our study include the fact that AST was not measured in more quadrants of the sclera. However, the measurement in the superior and inferior position at 12 and 6 o’clock was often not possible due to the reduced visibility of the sclera. Another limitation was the rather small sample size. In follow-up studies, scleral thickness and the rigidity of the sclera should be determined in a larger collective.

## 5. Conclusions

In conclusion, compared with the age- and refraction-matched controls, patients with CSC and PNV showed a significantly thicker anterior sclera. This study is the first to investigate scleral thickness in eyes with PNV. Our data indicate that a significantly thickened sclera may represent an ocular outflow resistance for the vortex veins in eyes with CSC and PNV, and secondarily cause a chronic overload of the venous system, congruent with the pathophysiology of UES. Chronic venous system backflow in the choroid may cause anterograde filling delay of the choriocapillaris with the formation of ischemic areas and may result in thickening of the Haller’s layer. These two factors combined may ultimately lead to the development of typical pachychoroid features.

## Figures and Tables

**Figure 1 jcm-12-03102-f001:**
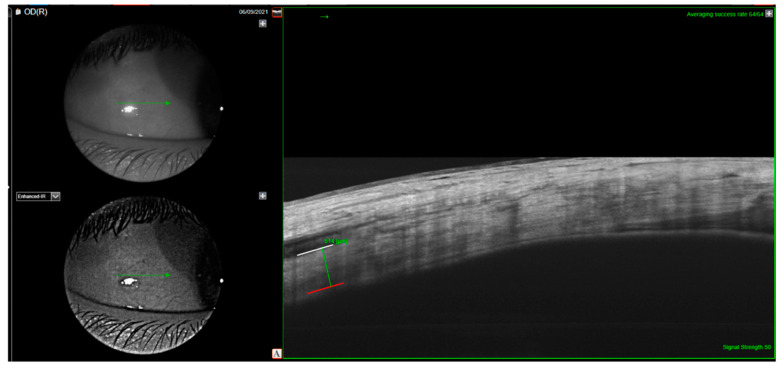
Anterior segment SS-OCT 6 mm scan. Measurement of AST (green vertical line) underneath the insertion site of the lateral rectus muscle using the built-in caliper function. The white horizontal line represents the anterior scleral border. The red line represents the posterior scleral border.

**Figure 2 jcm-12-03102-f002:**
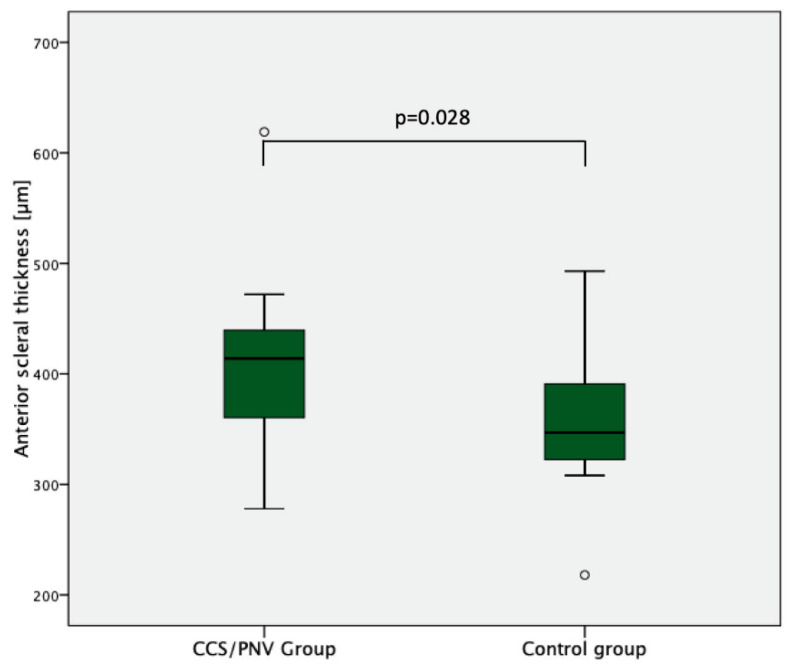
AST in the CSC/PNV and the control group. Boxplots showing significantly increased AST in the CSC/PNV group compared with the control group.

**Figure 3 jcm-12-03102-f003:**
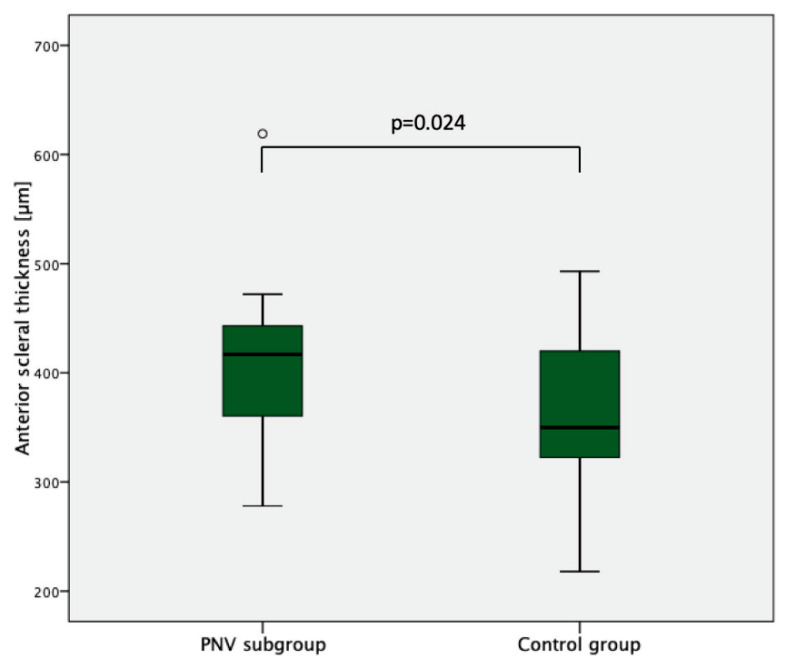
AST in the PNV subgroup and the control group. Boxplots showing significantly increased AST in the PNV subgroup compared with the control group.

**Table 1 jcm-12-03102-t001:** Overview of the patients’ treatment history in the CSC/PNV group. MSP: micropulse laser treatment; EPM: subthreshold laser treatment via endpoint management; PDT: photodynamic therapy; VEGF: vascular endothelial growth factor.

Patient No.	Age	Gender	Type of Pachychoroid Disease	Treatment History
1	37	M	Acute CSC	None
2	42	M	Acute CSC	Focal MSP
3	58	M	Chronic CSC	None
4	61	M	Chronic CSC	None
5	37	M	Chronic CSC	Spironolactone, 2 months
6	46	M	Chronic CSC	Spironolactone, 3 months
7	42	F	Chronic CSC	Spironolactone, 3 months
8	53	M	Chronic CSC	Spironolactone, 3 months
9	49	M	Chronic CSC	Spironolactone, 3 months
10	56	F	Chronic CSC	Spironolactone, 6 months
11	45	M	Chronic CSC	Spironolactone, 6 months
12	52	M	Chronic CSC	Spironolactone, 5 months, EPM
13	57	M	Chronic CSC	2xEPM
14	53	M	Chronic CSC	PDT
15	60	F	PNV	3 × anti-VEGF
16	61	F	PNV	3 × Anti-VEGF
17	51	F	PNV	3 × Anti-VEGF
18	63	M	PNV	6 × Anti-VEGF
19	55	F	PNV	8 × Anti-VEGF
20	53	M	PNV	9 × Anti-VEGF
21	66	F	PNV	37 × Anti-VEGF
22	59	F	PNV	25 × Anti-VEGF, Spironolactone, 3 months
23	51	M	PNV	32 × Anti-VEGF, 2 × PDT

**Table 2 jcm-12-03102-t002:** Demographic characteristics of the CSC/PNV group compared with the control and the distribution of pachychoroidal features in the CSC/PNV group.

	CSC/PNV Group (Mean ± SD; Range)	Control Group (Mean ± SD; Range)	*p*
No. of eyes (n) No. of patients (n)	23 23	23 23	
Gender (m/f)	15 m/8 f	14 m/9 f	
Mean age (y)	51.5 ± 8.0 (36 to 65)	47.0 ± 12.9 (30 to 83)	*p* = 0.122
SE	0.60 ± 2.23 (−5.5 to 4.6)	−0.93 ± 2.45 (−4.9 to 5.8)	*p* = 0.441
Pachychoroid type			
Acute CSC (n)	3		
Chronic CSC (n)	11		
PNV (n)	9		
Anterior scleral thickness (µm)	403.5 ± 68.6 (278 to 619)	362.5 ± 62.6 (218 to 493)	*p* = 0.028
Subfoveal choroidal thickness (µm)	392.8 ± 92.8 (191–523)	330.95 ± 116.5 (167–609)	*p* = 0.04

SD-OCT image analysis.

## Data Availability

Data available on request due to privacy restrictions.

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
