# Peer review of "Scleral Thickness as a Risk Factor for Central Serous Chorioretinopathy and Pachychoroid Neovasculopathy"

_jcm, 2023, doi:10.3390/jcm12093102_

Round 1

Reviewer 1 Report

The authors present an interesting manuscript on anterior scleral thickness in PNV. 

The manuscript is well written. 

However, there are a few concerns:

Although the difference in means was significant, there are several overlapping values of AST between PNV and normal control. How would the authors interpret this in a practical scenario?

Also, the mean spherical equivalent (SE) of control eyes is seen to be myopic compared to hyperopic SE in the PNV group. This could have resulted in the inclusion of more myopic eyes with thinner sclera in the control group. Could this have resulted in bias in results and interpretation? 

Page 5 line 139: "AST showed a tendency to increase in thickness with increasing hyperopia (R=0.23, p=0.30 and R=0.21, p=0.33). " It would not be right to make this statement with nonsignificant p-values. kindly rephrase.

Page 5 line 145: "Despite anti-VEGF treatment, the PNV subgroup did not show a significant thinning of AST compared to the CSC group (403.5 ± 46.2 (341-472) μm vs. 408.4 ± 81.4 (278-619) μm, p=0.75)."  Why do the authors think that it would get influenced by anti-VEGF injection? Were the injection site and measurement sites the same?

Author Response

               Munich, April 2023

Dear Chief Editor Prof. DeAngelis, dear reviewers,

We would like to thank you for your valuable comments and detailed suggestions. We found them very helpful in revising our manuscript. We reply in detail to each of your comments below on a point-to-point basis and hope that our revised manuscript is acceptable for publication in the renowned Journal of Clinical Medicine.

Submission ID jcm-2313718

Title: " Scleral thickness as a risk factor for central serous chorioretinopathy and pachychoroid neovasculopathy "

With best regards from Munich

Dr. Leonie F. Keidel on behalf of the entire group

Reviewer 1

  1. Although the difference in means was significant, there are several overlapping values of AST between PNV and normal control. How would the authors interpret this in a practical scenario?

Response 1: Thank you for your profound revision. You are right, there are overlaps between the two groups, yet AST showed to be in the upper range >350µm in every patient with PNV, compared to normal controls, where the majority of patients showed values below 350µm.

  1. Also, the mean spherical equivalent (SE) of control eyes is seen to be myopic compared to hyperopic SE in the PNV group. This could have resulted in the inclusion of more myopic eyes with thinner sclera in the control group. Could this have resulted in bias in results and interpretation?

Response 2: Thank you for pointing this out. We have also had these thoughts, yet since this difference is not significant, we do not think that this had a significant impact on the results of the evaluation. Nevertheless, in order to draw more precise conclusions, it is essential to evaluate a larger cohort of eyes in future studies.

  1. Page 5 line 139: "AST showed a tendency to increase in thickness with increasing hyperopia (R=0.23, p=0.30 and R=0.21, p=0.33). " It would not be right to make this statement with nonsignificant p-values. kindly rephrase.

Response 3: You are right. I rephrased the sentence to: “In the CSC collective as well as in the normal control group, the AST increased in thickness with increasing hyperopia (R=0.23, p=0.30 and R=0.21, p=0.33), yet these findings showed to be nonsignificant.”  

  1. Page 5 line 145: "Despite anti-VEGF treatment, the PNV subgroup did not show a significant thinning of AST compared to the CSC group (403.5 ± 46.2 (341-472) μm vs. 408.4 ± 81.4 (278-619) μm, p=0.75)." Why do the authors think that it would get influenced by anti-VEGF injection? Were the injection site and measurement sites the same?

Response 4: The measurement site and injection site were not the same, yet Zinkernagl et al.1 stated a generalized thinning of the sclera after anti-VEGF injections, which was nonsignificant, but still present. For this reason, we have added this sentence.

  1. Zinkernagel MS, Schorno P, Ebneter A, et al. Scleral thinning after repeated intravitreal injections of antivascular endothelial growth factor agents in the same quadrant. Invest Ophthalmol Vis Sci 2015;56(3):1894-900. doi: 10.1167/iovs.14-16204 [published Online First: 2015/02/26]

Reviewer 2 Report

The aim of the manuscript entitled Scleral thickness as a risk factor for central serous chorioretinopathy and pachychoroid neovasculopathy “ by Leonie F Keidel et al. was to investigate anterior scleral thickness (AST) as a risk factor for pachychoroid disorders. The scleral thickness in patients with CSC and PNV was assessed.The paper is generally well written.  The manuscript is correct in terms of the structure and the meritsThe Abstract and  the Introduction are clear and provide all the needed information for the readers. The statistical analysis is performed appropriately.  However, there are several issues that should be addressed before making a final decision.

 1.    Anterior scleral thickness is the main parameter investigated in the study, but the Authors did not mention AST in the fellow eye, was it assessed? Additionally, the study did not mention if pachychoroid was observed only in the eyes with CSC/PCV or in the fellow eye as well? Add information about the condition of the fellow eye in the Material section or Table 2.

2.    Regarding figure 1 -  it would be helpful to mark the eye structures on the OCT scan to clearly explain how the measurements were done. Similarily as in the available literature (Sawaguchi S, Terao N, Imanaga N, Wakugawa S, Tamashiro T, Yamauchi Y, Koizumi H. Scleral Thickness in Steroid-Induced Central Serous Chorioretinopathy. Ophthalmol Sci. 2022 Feb 8;2(2):100124. doi: 10.1016/j.xops.2022.100124 - Figure 1).

3.    In line 139-140 the Authors mentioned that “AST showed tendency to increase in thickness with increasing hyperopia”. Can you specify how many myopic, emmetropic and hyperopic patients were included into both groups (CSC/PNV and control)?  Insert this information in the Material section.

4.    Please change CCS into CSC in Table 2 (CSC term was used in the whole manuscript). 

5.   Interesting results were reported in the report by Sawaguchi S, Terao N, Imanaga N, Wakugawa S, Tamashiro T, Yamauchi Y, Koizumi H. Scleral Thickness in Steroid-Induced Central Serous Chorioretinopathy. Ophthalmol Sci. 2022 Feb 8;2(2):100124. doi: 10.1016/j.xops.2022.100124. The Authors showed that the mean scleral thickness value of eyes with sCSC (steroid-induced CSC) was significantly thinner than that of eyes with iCSC (idiopathic CSC). They suggested that the sclera has less involvement in the pathogenesis of sCSC than in that of iCSC. Do you have data on your patients regarding the use of steroids? Or all of them had idiopathic CSC?  Refer please to this report in the Discussion.

Minor concerns:

Line 40: Please define OCT as “optical coherence tomography (OCT)”.

Line 41: “They conclude” is a present tense, please change it into past. In manuscript mainly past tense was used. Please check carefully if there are any other tense differences to correct.

Line 43-45: “This is supported by the fact that in eyes with CSC, a significantly more frequent formation of intervenous anastomosis between the outflow areas of the vortex veins was observed in response to a chronic overload of the venous system compared to a control collective.” This sentence is too long and difficult to follow. Consider rephrasing it  as follows: “The Authors’ thesis is supported by the fact that in eyes with CSC chronic overload of the venous system was observed compared to a control collective. This excess in the outflow areas of the vortex veins was observed in response to formation of intervenous anastomosis and was significantly increased in eyes with CSC.”

Line 51: PNV was previously explained in line 32, please remove explanation.

Line 52: Please change “This study is the first to investigate scleral thickness also in eyes with PNV.” into “This study is the first to investigate scleral thickness  in eyes with PNV as well”

Line 54: Please change “For this prospective case-control study, 23 consecutive eyes presenting with CSC or PNV at the Hospital of the Ludwig Maximilians-University Munich, Department of Ophthalmology, Germany from March 2022 to June 2022 were included into this study” into “This prospective case-control study enrolled 23 consecutive eyes presenting with CSC or PNV at the Hospital of the Ludwig Maximilians-University Munich, Department of Ophthalmology, Germany from March 2022 to June 2022.  

Line 56-57: Please change “These were then matched” into “They were matched” 

Line 57: Please explain spherical equivalent as SE.

Line 65: Please change “Autorefractometer” into “Auto refractometer”.

Line 82: Define abbreviation AST  in the text in the first mention.

Line 82: Please write abbreviation of swept-source OCT (SS-OCT), it is then used in manuscript as abbreviation without explanation.

Line 157: Please remove “so-called”.

Line 158 and then 160, 172, 195: Please correct “Haller layer/veins” into “Haller’s layer/veins”. Please check if there are any other errors in nomenclature.

Line 161: Please explain RPE and then remove its explanation in line 162-163.

Line 191: Please remove “also”.

Line 233: Please remove “also”

Line 236-239: Please correct sentence: “A chronic backflow into the venous system of the choroid may, on the one hand, cause anterograde filling delay of the choriocapillaris with formation of ischemic areas and, on the other hand, may result in thickening of the Haller layer.” 

Consider rephrasing it  as follows: “Chronic venous system backflow in the choroid may cause anterograde filling delay of the choriocapillaris with formation of ischemic areas as well as may result in thickening of the Haller’s layer.” English should be revised by a native speaker

Author Response

               Munich, April 2023

Dear Chief Editor Prof. DeAngelis, dear reviewers,

We would like to thank you for your valuable comments and detailed suggestions. We found them very helpful in revising our manuscript. We reply in detail to each of your comments below on a point-to-point basis and hope that our revised manuscript is acceptable for publication in the renowned Journal of Clinical Medicine.

Submission ID jcm-2313718

Title: " Scleral thickness as a risk factor for central serous chorioretinopathy and pachychoroid neovasculopathy "

With best regards from Munich

Dr. Leonie F. Keidel on behalf of the entire group

Reviewer 2

  1. Anterior scleral thickness is the main parameter investigated in the study, but the Authors did not mention AST in the fellow eye, was it assessed? Additionally, the study did not mention if pachychoroid was observed only in the eyes with CSC/PCV or in the fellow eye as well? Add information about the condition of the fellow eye in the Material section or Table 2.

Response 1: Thank you for your profound and detailed revision, to help us improve our manuscript. All patients showed pachychoroid in the fellow eye, yet to allow unbiased evaluation, we included only one eye per patient in the statistical analysis. The AST of the fellow eye was not assessed in the study. I included information on the SFCT of the fellow eyes to the manuscript.:

“CSC/PNV partner eyes all showed a pachychoroid (mean SFCT: 405.4 ± 120.7 (260-564) µm). There were eight eyes with PPE and 2 eyes with chronic CSC.”

  1. Regarding figure 1 - it would be helpful to mark the eye structures on the OCT scan to clearly explain how the measurements were done. Similarily as in the available literature (Sawaguchi S, Terao N, Imanaga N, Wakugawa S, Tamashiro T, Yamauchi Y, Koizumi H. Scleral Thickness in Steroid-Induced Central Serous Chorioretinopathy. Ophthalmol Sci. 2022 Feb 8;2(2):100124. doi: 10.1016/j.xops.2022.100124 - Figure 1).

Response 2: Thank you for this good advice. I marked the eye structure, as it was done so well in the manuscript by Sawaguchi et al. and adjusted the heading.

  1. In line 139-140 the Authors mentioned that “AST showed tendency to increase in thickness with increasing hyperopia”. Can you specify how many myopic, emmetropic and hyperopic patients were included into both groups (CSC/PNV and control)? Insert this information in the Material section.

Response 3: This is a good idea to further specify the refraction. I inserted information on refraction of all of the groups next to the details on their respective SEs in the patient characteristics section. For the CSC/PNV group I stated: “10 patients had a myopic refraction (2 PNV patients), 13 patients had a hyperopic refraction.” For the normal control group I stated: “14 patients had a myopic refraction, 3 were emmetropic and 6 patients were hyperopic.”

  1. Please change CCS into CSC in Table 2 (CSC term was used in the whole manuscript).

Response 4: Thank you for pointing out this error in abbreviation. I changed the abbreviations in table 2.

  1. Interesting results were reported in the report by Sawaguchi S, Terao N, Imanaga N, Wakugawa S, Tamashiro T, Yamauchi Y, Koizumi H. Scleral Thickness in Steroid-Induced Central Serous Chorioretinopathy. Ophthalmol Sci. 2022 Feb 8;2(2):100124. doi: 10.1016/j.xops.2022.100124. The Authors showed that the mean scleral thickness value of eyes with sCSC (steroid-induced CSC) was significantly thinner than that of eyes with iCSC (idiopathic CSC). They suggested that the sclera has less involvement in the pathogenesis of sCSC than in that of iCSC. Do you have data on your patients regarding the use of steroids? Or all of them had idiopathic CSC? Refer please to this report in the Discussion.

Response 5: Thank you for pointing out this brilliant manuscript by Sawaguchi et al. Only 4 of the patients used steroids beforehand. I looked closer on the AST of these patients and added this phrase to the results:

“The CSC/PNV was idiopathic in 19 patients (iCSC), whereas in 4 patients a steroid-induced CSC (sCSC) was present.”

“AST in patients with sCSC (341.8 ± 51.9 (278-392) µm) was remarkably low compared to those with iCSC (420.2 ± 64.6 (322-619 µm).”

And I added these phrases and the publication to the discussion:

“In a recent important work by Sawaguchi, Koizumi et al. mean AST in eyes with sCSC was shown to be significantly thinner than in eyes with iCSC. Our findings go hand in hand with this work.”

  1. Line 40: Please define OCT as “optical coherence tomography (OCT)”.

Response 6: Thank you for showing us that in this case the abbreviation has not yet been mentioned. I have adjusted this accordingly.

  1. Line 41: “They conclude” is a present tense, please change it into past. In manuscript mainly past tense was used. Please check carefully if there are any other tense differences to correct.

Response 7: Thank you for pointing this out. I corrected the verb into the past tense and revised the manuscript. Likewise, I went through the manuscript again for the occurrence of a present tense.  I could not find any other incorrect tenses.

  1. Line 43-45: “This is supported by the fact that in eyes with CSC, a significantly more frequent formation of intervenous anastomosis between the outflow areas of the vortex veins was observed in response to a chronic overload of the venous system compared to a control collective.” This sentence is too long and difficult to follow. Consider rephrasing it as follows: “The Authors’ thesis is supported by the fact that in eyes with CSC chronic overload of the venous system was observed compared to a control collective. This excess in the outflow areas of the vortex veins was observed in response to formation of intervenous anastomosis and was significantly increased in eyes with CSC.”

Response 8: Thank you for rephrasing this thought in such a perfect way and for pointing out that the sentence had been hard to understand. I changed the sentence into your rephrased sentence.

  1. Line 51: PNV was previously explained in line 32, please remove explanation.

Response 9: You are right, I removed the explanation.

  1. Line 52: Please change “This study is the first to investigate scleral thickness also in eyes with PNV.” into “This study is the first to investigate scleral thickness in eyes with PNV as well”

Response 10: I changed the sentence.

  1. Line 54: Please change “For this prospective case-control study, 23 consecutive eyes presenting with CSC or PNV at the Hospital of the Ludwig Maximilians-University Munich, Department of Ophthalmology, Germany from March 2022 to June 2022 were included into this study” into “This prospective case-control study enrolled 23 consecutive eyes presenting with CSC or PNV at the Hospital of the Ludwig Maximilians-University Munich, Department of Ophthalmology, Germany from March 2022 to June 2022.

Response 11: Thank you for this rebuild in sentence structure. It improves the beginning of the Materials section.

  1. Line 56-57: Please change “These were then matched” into “They were matched”

Response 12: I changed the phrase.

  1. Line 57: Please explain spherical equivalent as SE.

Response 13: Good advice. I added the abbreviation.

  1. Line 65: Please change “Autorefractometer” into “Auto refractometer”.

Response 14: Auto refractometer is already spelled like this in the manuscript.

  1. Line 82: Define abbreviation AST in the text in the first mention.

Response 15: Thank you for pointing this out. I overlooked that the first statement was in the abstract and not in the text. I defined it in the text accordingly.

  1. Line 82: Please write abbreviation of swept-source OCT (SS-OCT), it is then used in manuscript as abbreviation without explanation.

Response 16: Thank you for pointing this out as well. I overlooked that the first statement was in the abstract and not in the text. I defined it in the text accordingly.

  1. Line 157: Please remove “so-called”.

Response 17: I removed the phrase.

  1. Line 158 and then 160, 172, 195: Please correct “Haller layer/veins” into “Haller’s layer/veins”. Please check if there are any other errors in nomenclature.

Response 18: Thank you for pointing out this misspelling. I corrected Sattler layer into Sattler’s layer accordingly.

  1. Line 161: Please explain RPE and then remove its explanation in line 162-163.

Response 19: You are right. I added an explanation of RPE earlier in line 161 and then removed the explanation in line 162.

  1. Line 191: Please remove “also”.

Response 20: I removed “also”.

  1. Line 233: Please remove “also”

Response 21: I removed “also”.

  1. Line 236-239: Please correct sentence: “A chronic backflow into the venous system of the choroid may, on the one hand, cause anterograde filling delay of the choriocapillaris with formation of ischemic areas and, on the other hand, may result in thickening of the Haller layer.” Consider rephrasing it as follows: “Chronic venous system backflow in the choroid may cause anterograde filling delay of the choriocapillaris with formation of ischemic areas as well as may result in thickening of the Haller’s layer.” English should be revised by a native speaker.

Response 22: Thank you for rephrasing the sentence. I have replaced the sentence for your sentence. A native speaker revised our manuscript.

Round 2

Reviewer 1 Report

Queries have been well addressed. There are no further comments.

Author Response

Munich, April 2023

Dear Chief Editor Prof. DeAngelis, dear reviewers,

We would like to thank you for the second revision and further valuable comments. We reply in detail to each of your comments below on a point-to-point basis and hope that our revised manuscript is acceptable for publication in the renowned Journal of Clinical Medicine.

Submission ID jcm-2313718

Title: " Scleral thickness as a risk factor for central serous chorioretinopathy and pachychoroid neovasculopathy "

With best regards from Munich

Dr. Leonie F. Keidel on behalf of the entire group

Reviewer 1

  1. Queries have been well addressed. There are no further comments.
    Response 1: Thank you for your profound and detailed revision, to improve our manuscript.

Reviewer 2 Report

Dear Authors,

Thank you for the opportunity to review this revised manuscript. The changes has made the study more interesting and easier to understand for the general audience.

Analysis of the revised manuscript has shown several minor errors that need to be corrected:

Line 72:

Define abbreviation OCT.

Line 85 

Please add a dot “.” at the end of sentence.

Line 115

Indocyanine green angiography was firstly used without abbreviation - please add “(ICGA)”.

Line 210

Normal group = control group, please remove “normal” or “control” from this sentence. 

Author Response

                                                                                                                         Munich, April 2023

Dear Chief Editor Prof. DeAngelis, dear reviewers,

We would like to thank you for the second revision and further valuable comments. We reply in detail to each of your comments below on a point-to-point basis and hope that our revised manuscript is acceptable for publication in the renowned Journal of Clinical Medicine.

Submission ID jcm-2313718

Title: " Scleral thickness as a risk factor for central serous chorioretinopathy and pachychoroid neovasculopathy "

With best regards from Munich

Dr. Leonie F. Keidel on behalf of the entire group

Reviewer 2

  1. Line 72. Define abbreviation OCT.

Response 1: Thank you for your second revision, to help us improve our manuscript and to correct remaining errors. I defined OCT.

  1. Line 85. Please add a dot “.” at the end of sentence.

Response 2: I am sorry for the missing of the punctuation. I added the dot.

  1. Line 115. Indocyanine green angiography was firstly used without abbreviation - please add “(ICGA)”.

Response 3: Thank you for the hint. I added ICGA.

  1. Line 210. Normal group = control group, please remove “normal” or “control” from this sentence.

Response 4: Thank you for pointing out this error. I left “control” group.
